


# Chaotic Signatures and Global Solar Radiation model estimate over Nigeria, a Tropical region

*Adedayo Adelakun*[a*] and *Folasade Adelakun*[b]
[a]Department of Physics, Federal University of Technology, Akure, Ondo state, Nigeria
[b]Department of Crop, Soil and Pest, Federal University of Technology, Akure, Ondo state, Nigeria
*Corresponding author email address: d_onescientist@yahoo.com

**Abstract**

In a tropical region like Nigeria, accurate estimation and chaotic signatures of global solar radiation ($R_s$) are essential to the design of solar energy utilization systems in PV technology companies and one conservative energy source required in developing drying devices in today's mechanized Agriculture. The $R_s$ model is a function of solar declination angle, temperature difference, and relative humidity. In this paper, the daily re-analyzed atmospheric data obtained from the archive of ERA-Interim was used to estimate the nonlinear Global Solar radiation model and investigated chaotic signatures across the tropical climatic regions of Nigeria. The well-known statistical tools were used to analyze the chosen meteorological parameters and the correlation was found to be perfect, close with low values of RMSE across the selected regions over Nigeria. For proper modeling and prediction of the underlying dynamics, the extensive chaotic measures of phase space reconstruction using recurrence plots and recurrence quantification analyses are also presented, analyzed and discussed with the appropriate choice of embedded dimension, m, and time delay $\tau$.

**Keywords**: Recurrence Plot (RP); Recurrence Quantification Analyses (RQA); Phase space reconstruction (PSR); Global Solar Radiation; Chaotic Signatures;Embedded Dimension; Time Delay

The radiant energy from the sun is one of the most available and renewable resources across the season in a tropical region like Nigeria. The information, therefore, suggests how vital the solar irradiance can be useful in Agriculture and Photovoltaic technology companies. Based on the scarcely gauged of global solar radiation (GSR) at meteorological stations in developing countries. This demand necessitates a better understanding of the underlying dynamics for better prediction mostly by the nonlinear Global Solar radiation model estimate and chaotic signature measurement. The optimum usage of meteorological parameters such as solar radiation, relative humidity and temperature difference needs further studies, using RPs and RQA measures. However, several data such as rainfall data, geomagnetic data, ionospheric data, wind speed data etc obtained from different parts of the world have been estimated with several models and applied to RQA measures for better prediction and modeling. Using RPs and RQA, features due to external effects such as harmattan and intertropical discontinuity (ITD) on solar radiation data in this tropical region were uniquely identified. Meanwhile, the inverse characteristic behavior of solar radiation and relative humidity were vividly maintained. The results show a very low value of RMSE while the value of $R^2$ is very closed to 1, which depicts a good prediction for all locations. However, the highest values of both SSE and RMSE, as well as the lowest value of $R^2$ were observed in kano station, which indicates high solar irradiance location. The RPs reviewed the observed clusters points around the parallel diagonal lines with short segments, which implies the presence of chaos. Additional complex measure, the RQA also shows that the solar radiation during



the dry season of the months has lower values of Lmax, determinism and entropy, and higher values during the wet season of the months.

# 1    Introduction

Solar energy has proven to be one of the clean and most harnessed renewable energy sources in the world. It has been singled out as the only source that can sustain the earth and greatly influence nonlinear conditions of weather and climate (Govindasamy and Chetty 2018). The earth receives about 174 Petawatts (i.e maximum of 75%) of sunlight insolation on the Earth's surface which is almost 10,000 times the total amount of energy used by humans on Earth, as taken from sources such as, oil, coal, natural gas, nuclear and hydroelectric power combined (Rhodes 2010; Agbo and Oparaku 2006). Even though the energy source account for reflection, absorption and atmospheric transmittance which are mainly caused by cloudy atmospheric conditions such as complex internal activities of aerosols, clouds and gas molecules. The source has been capable of reducing climate and weather events such as the greenhouse effect, global warming impacts and natural hazards (Uckan and Khudhur 2018).

Some references have reported the problem of insufficient solar radiation measurement due to the high cost, maintenance, calibration of measuring devices among other factors which have been the major setback in maximizing the usage of the energy source (Salisu 2017; Sarker and Sifat 2016; Almorox and Fernandez 2004; Almorox *et al.* 2011). Meanwhile, to create an alternative means for better modeling and prediction of solar energy in even areas with no or less meteorological data, the transition from irradiation measurement to data analysis is necessary. In 2015, Kutty *et al* proposed three methods of collecting solar radiation data: (i) direct estimation through in situ measurements (2) satellite data and indirect estimation and (3) statistical techniques. The use of a numerical approach for solar radiation estimate becomes a useful alternative for adequate information with different reports. For instance, the numerical and empirical models have also been developed for different meteorological parameters such as temperature, sunshine hours, relative humidity, longitude, latitude, altitude and sea level pressure (Saeed *et al.* 2019). In the same vein, several investigations on linear and nonlinear global solar radiation estimations model within and outside Nigeria have been carried-out (Angstrom 1924; Korachagaon *et al.* 2015; Ajayi *et al.* 2014; Akpootu and Sanusi 2013; Ogunjo *et al.* 2015; Hassan *et al.* 2011; Ahmad and Tiwari 2011). Also, the application of Local global solar radiation data which is largely required for architectural design, evapotranspiration estimates, irrigation, has been reported (Sarker and Sifat 2016).

Since the nonlinear approach to atmospheric convection using Lorenz's model (Lorenz 1963), various atmospheric situations have been applied to the field of science and engineering. These numerous scientific disciplines help in providing the information that is needed in forecasting the weather condition. That is, the complexity of the global solar radiation and the inherent irregularities occurrence in space can be identified to be chaotic or hyperchaotic based on the availability of climatological data which has been enriched with the theory of nonlinear dynamics. It is worth noting that large time-series data is very vital in providing more insight into the internal activities of the atmosphere (Kantz and Schreiber 2003). For example, a complex situation in the ionosphere (Rabiu *et al.* 2015), global radio link fading (Adelakun *et al.* 2019; Ojo *et al.* 2019) and wind speed data for the optimization of power generation (Adeniji *et al.* 2018) have been recently reported. Gan *et al.* (2012) further tested for nonlinearity in solar radiation data using the method of fast surrogate test and revealed. Other tests based on Neuro-fuzzy approach (Omid *et al.* 2012), hidden Markov models (Hocaoglu 2011), artificial neural network (Sozen *et al.* 2004), swarm-optimized neural network (Lazzus 2011), autoregressive integrated moving average (ARIMA) (Wu and Chee 2011), and parametric models (Katiyar and Pandey 2013) on solar irradiance has been extensively discussed. Recently, Ogunjo *et al.* (2015)



measured the chaotic features in Akure, a tropical station in South-western Nigeria based on half-
hourly and daily data. So far, various nonlinear global radiation models based on clearness index and
temperature difference air temperature have been proposed (Sarker and Sifat 2016). However, the
estimated model in this paper expressed the relationship between clearness index, relative humidity,
and temperature difference. We also employed the chaotic quantifiers, RP and RQA, to investigate
the daily and seasoning variations in the meteorological parameters for the four basic climatic regions
in Nigeria, namely; the Coastal region, Guinea savannah region, Midland region and Sahel savannah
region. This paper presents a nonlinear global solar radiation model estimate and chaotic signatures
from available meteorological data. Section 2 comprises of the study area, data analysis, and statisti-
10 cal performance evaluation. The mathematical analyses of the used chaotic quantifiers are discussed
in section 3, while section 4 concludes the paper.

## 2 Methodology

### 2.1 Study area and Data Analysis

The re-analyzed atmospheric data from four climatic regions of Nigeria were obtained from the archive
of the ERA-Interim database. Ten years of daily data covering the period of 2006 to 2015, for so-
lar radiation, relative humidity, and temperature differences were collected and analyzed for this
research. Table 1 shows the selected geographical locations includes Kano (Sahel savannah region),
South-Western station, Akure (Coastal region), Ilorin (Midland region) and Yola (Guinea Savannah).

Table 1: Regions and their Geographical location

|  |  | Geographical Location | |
| --- | --- | --- | --- |
| Region | Station | Latitude ($^oN$) | Logitude ($^oE$) |
| Sahel savannah | Kano | 12.00 | 8.59 |
| Guinea savannah | Yola | 9.20 | 12.50 |
| Midland | Ilorin | 8.50 | 4.55 |
| Coastal | Akure | 7.25 | 5.20 |

The collected data were used in global solar nonlinear model estimation, given as:

$$R_s = a(1 + bH)(1 - \exp(-c\Delta T^n)) \tag{1}$$

which can be expressed as,

$$R = a(1 + bH)(1 - \exp(-c\Delta T^n)) \times R_a \tag{2}$$

where clearness index, $R_s = R/R_a$, is the solar ratio which depicts the ratio of the global solar ra-
diation (R) to the extraterrestrial radiation, $R_a$, H is the Relative Humidity and $\Delta T$ is the differences
between maximum and minimum temperatures. The estimated constant parameters a, b, c and n are
dimensionless parameter estimates to be determined by least square method. $R_s$ is the dependent
variable and the other two non-dependent variables are given as $\Delta T(K)$ and H (%), respectively. The
extraterrestrial radiation ($R_a$) was calculated from the expression (Duffie and Beckman 2006):

$$R_a = \frac{24 \times 60}{\pi} I_{sc} E_o(\omega_s sin\varphi sin\delta + cos\varphi cos\delta sin\omega_s) \tag{3}$$

$R_a$ = extraterrestrial radiation $[MJm^{-2}day^{-1}]$, $I_{sc}$ = solar constant =1353 $W\Delta m^2$, $E_o$ = inverse
relative distance Earth-Sun, $\omega_s$ = sunset hour angle[rad], $\varphi$ = latitude of the site [rad], $\delta$ = solar
declination [rad], J is the Julian calendar day of the year (January 1st corresponds to J = 1, December
31st corresponds to J = 365).





1 The inverse relative distance Earth-Sun, $E_o$, and the solar declination, $\delta$, are given by (Hassan *et*
2 *al.* 2006; Chang 2010):

$$E_o = 1 + 0.033 cos(\frac{2\pi}{365}J) \tag{4}$$

$$\delta = 0.409 sin(\frac{2\pi}{365}J - 1.39) \tag{5}$$

3 The sunset hour angle, $\omega_s$, is given by:

$$\omega_s = \frac{\pi}{2} - arctan(\frac{-tan\varphi tan\delta}{X^{0.5}}) \tag{6}$$

4 where

$$X = 1 - (tanj)^2(tan\delta)^2 \tag{7}$$

## 5 2.2 Statistical Performance Evaluation

Fit measured is very important in comparing and assessing models with the aid of statistical tools.
The statistical tools have been used in recent years extensively to evaluate different models ranging
from linear, quadratic, third-degree, logarithmic and exponential models after comparing and assess-
ing (Uckan and Khudhur 2018). In this work, $R^2$, SSE, and RMSE are the performance indicators
used for regression analysis to evaluate the proposed nonlinear model. Besides, the RMSE test also
provides information on the short-term evaluation by allowing a term by term comparison of actual
deviation between the calculated value and the measured value i.e. estimates the concentration of
the data around the fitted equation. Low RMSE values indicate that the model accurately represents
the observed global solar irradiance. Meanwhile, the SSE measures how far the data are from the
model's predicted values, while $R^2$ is used to determine the performance of a model in terms of its
suitability. This value is one of the most significant indicators for comparing models because it is
dimensionless and easily calculated. Ideally, a model is considered to be perfect if $R^2 = 1$. This value
indicates that the estimated values match perfectly with the observed values. The $R^2$ is the square of
the correlation between the predictor and response which signifies good fit if close to 1 which depicts
good prediction and weak if close to zero. That is, $R^2=1$ is an indication of a perfect match between
the predicted and the observed values. The expression for $R^2$ is:

$$R^2 = 1 - \frac{\sum_{i=1}^{N}[\sum(I_{i,obs} - I_{i,pre})^2]}{\sum_{i=1}^{N}[\sum(I_{i,obs} - \bar{I_{i,obs}})^2]} \tag{8}$$

The RMSE is however defined as:

$$RMSE = \frac{100}{G_m}\sqrt{\frac{\sum(I_{i,pre} - I_{i,obs})^2}{N}} \times 100\% \tag{9}$$

while SSE can also be expressed as:

$$SSE = \sum_{i=1}^{N}[\sum(I_{i,pre} - I_{i,obs})^2] \times 100\% \tag{10}$$

It is worth noting that the RMSE is always positive while a zero value is ideal. In the Eqs. (8-10),
N indicate the total number of observations, $G_m$ is the mean of N measured values, $I_{i,pre}$ is the $i^{th}$
predicted value, $I_{i,obs}$ is the $i^{th}$ observed value. The mean of the observed value is given as $\bar{I_{i,obs}}$.





## 3 Chaotic Quantifiers

In nonlinear science, a PSR can be used in estimating the characteristic properties of a natural system such as global solar radiation. It requires decision making regarding the size of the space, the value of the time shifts between the coordinates, and another important—although often overlooked—aspect, that is, which one or which combination of observable(s), if several of them are available will be used for the reconstruction (Fraser and Swinney 1986; Kennel *et al.* 1992). The multidirectional aspect of the data can be revealed based on the reconstruction of average phase portraits (Rabiu *et al.* 2015; Takens 1981) in which the embedding parameters, the dimension m, and the delay $\tau$ must be carefully chosen. Meanwhile, the choice of embedding is very important in other to determine the phase space trajectory of solar irradiance. However, the minimum embedding dimension can be determined using false nearest neighbor method (Unnikrishnan 2010; Unnikrishnan and Ravindran 2010). That is, the smallest sufficient or minimum embedding dimension is required in PSR from the observed data for which the time delay must be obtained (Kantz and Schreiber 2003; Fraser and Swinney 1986; Kennel *et al.* 1992). The approach includes:

(i) Computation of some invariant measure on the reconstructed attractor, which will change if the current embedding dimension is too small, but persist for large value. This method is subjective and required a lengthy data set.
(ii) Investigation of the changes in the neighborhood of phase space points may be applied if there are changes to the value of embedding dimensions. Although, inappropriate embedding dimensions can affect or cause an increase in the amount of FNN.
(iii) The single value decomposition of an initial set of PSR vectors also reveals the smallest number of uncorrelated directions in phase space, which can be used as an embedded dimension.

In accordance to Takens theorem (Takens 1981), the PSR is defined as

$$Y(i) = [x(i), x(i+\tau), x(i+2\tau), ...., x(i+(m-1)\tau)], i = 1, 2, ..., N; (N = n - (m-1)\tau) \qquad (11)$$

where, $x_1$, $x_2$, $x_3$....., $x_{n-1}$, $x_n$ denotes the chaotic time series, embedding dimension, $m$, time delay, $\tau$ and N is the number of samples after reconstruction.
In the same vein, Fraser and Swinney (1986) also proposed a method of AMI to determine the delay, and the best choice for the delay is where the AMI has its smallest local minimum. However, the delay, $\tau$ has to be carefully chosen due to linear dependence between the subsequent vectors which impregnated from random errors and low measurement precision. The AMI is expressed as:

$$I(\tau) = -\sum_{\psi,\phi} P\psi, \phi(\tau) log \frac{P_{\psi,\phi}(\tau)}{P_\psi P_\phi} = \left\langle log \frac{P_{u_i}, u_{i+\tau}}{P_{u_i} P_{u_{i+\tau}}} \right\rangle \qquad (12)$$

where $P_{\psi,\phi}(\tau)$ is the joint probability that $u_i = \psi$ and $u_{i+\tau} = \phi$. $P_\psi$ and $P_\phi$ are the probability that $u_i$ has the value $\psi$ and $\phi$, respectively. Also, the generalized mutual information for higher dimensional joint distributions
$P_{u_i}, u_{i+\tau}, ..., u_{i+(m-1)\tau}$ can be defined inform of redundancy, $R^m(\tau)$ as (Kennel *et al.* 1992):

$$R^m(\tau) = \left\langle log \frac{P_{u_i, u_{i+\tau}}, ..., u_{i+(m-1)\tau}}{P_{u_i}, P_{u_{i+\tau}}, ..., P_{u_{i+(m-1)\tau}}} \right\rangle \qquad (13)$$

Likewise the marginal redundancy is given as (Fraser and Swinney 1986):



$$R^m(\tau) = R^{m+1}(\tau) - R^m(\tau) \tag{14}$$

On the other hand, one of the chaotic quantifiers necessary in the study of the nonlinear behavior or complex signature of any dynamical system is Lyapunov exponent. This quantifier determines the level of chaos in natural systems using time series data point (Unnikrishnan 2010; Unnikrishnan and Ravindran 2010). A positive Lyapunov exponent indicates divergence of trajectory in one dimension, or an expansion of volume, which can also be said to indicate repulsion or attraction from a fixed point. A positive Lyapunov exponent is an indication or evidence of chaos in a dissipative deterministic system i.e. the positive Lyapunov exponent indicates divergence of trajectory in one direction or expansion of value, and a negative value shows convergence at trajectory or contraction of volume along another direction. According to Wolf *et al.* (1985), the rate of divergence of any trajectory of any dynamic systems rests lonely on LLE. The Lyapunov exponent from the different location was computed in this work, by scanning the state space of the solar radiation basically for the entire four regions with $\tau = 10$ and m=7, respectively (Rosenstein *et al.* 1993; Hegger *et al.* 1994). The Lyapunov exponent $\lambda$ can be expressed as

$$\lambda = \frac{1}{t} \ln \frac{\triangle x(t)}{x(0)} = \frac{1}{t} \sum_{i=1} \ln(\frac{\triangle x(t)}{\triangle x(t_i - 1)}) \tag{15}$$

It is worth noting that entropy is reciprocal to Lyapunov exponent and it is mostly used in both physics and information theory to describe the amount of uncertainty or information inherent in an object or system (Kantz and Schreiber 2003). Tsallis entropy has been used extensively for various systems with complex signatures such as magnetospheric dynamics (Balasis *et al.* 2008; 2010), ionospheric dynamics (Ogunsua *et al.* 2014), tropospheric dynamics (Tsallis 1988; Boon and Tsallis 2003) among others. But the characterization of entropy by an index $q$, leads to nonextensive statistics. The parameter $q$ itself is not a measure of the complexity of the system, but measures the degree of non-extensivity of the system: $q \rightarrow 1$ corresponds to the standard extensive Boltzmann–Gibbs statistics which generalizes the Boltzmann–Gibbs theory. Also, it is the time variations of the Tsallis entropy for a given $q(S_q)$ that quantify the dynamic changes in the complexity of the system, that is, lower values of $S_q$ characterize the portions of the signal with lower complexity (Fraser 1989). The Tsallis entropy $S_q$ is calculated using

$$S_q = k \frac{1}{q-1} (1 - \sum_{i=1}^{W} P_i^q). \tag{16}$$

and the entropic index q for systems A and B, respectivley, characterizes the degree of nonadditivity reflected in the following pseudo-additivity rule:

$$S_q(A+B) = S(q_A) + S(q_B) + (1-q)S_q(A)S_q(B). \tag{17}$$

where $p_i$ is the probabilities associated with the microscopic configurations, W is their total number, $q$ is a real number, and k is Boltzmann's constant.

A comparison in this work shows the complex link between the Lyapunov exponent of the solar radiation and that of its Tsallis entropy. This is based on the fact that Tsallis entropy has been linked to a significant degree of response to the edge of chaos and chaotic regime dynamical systems due to its non-extensive nature (Baranger *et al.* 2002; Anastasiadis *et al.* 2005), and it has been linked to weak chaos and the vanishing LLE (Kalogeropoulos 2012; 2013]. The basis for comparison has been reported that the Lyapunov exponent varies directly as the Tsallis entropy (complexity) of a system, based on the variation of the entropy index q introduced by Tsallis and the nature of the system's dynamics. Similarly, the direct and indirect relationship of the temperature difference and relative humidity to solar radiation can also be confirmed.





In other to measure further the complexity in deterministic systems, Eckmann *et al.* (1987) introduced an RPs to show more insight into the optimal embedded dimension by visualizing the higher dimensional phase space through a 2D-representation. The close inspection of the recurrence structures has been developed and reported in Marwan *et al.* (2007) with several applications in physics, engineering, biology, and others. RPs exhibit a large scale pattern called typology, while the small scale pattern is texture. The typology pattern gives a global impression that is based on homogeneous, periodic, drift and disrupted. However, the texture pattern depends on closer inspection of single dots, diagonal lines, rectangular clusters of RPs. In other words, a single recurrence point contains no information about the state itself. However, an increase in the embedding dimension always cleans up single RPs by emphasizing the diagonal structures as diagonal lines. In RPs, when the amount of redundancy increases, the embedding dimension also increases which leads to distinct diagonal oriented structures. Therefore, RPs can be expressed in terms of trajectory $\vec{x_i}\ \varepsilon$ $\Re_n (i = 1, ..., N)$ in the n-dimensional phase space, which is the fundamental property of deterministic dynamical systems and is typical for nonlinear or chaotic systems (Argyris *et al.* 1994; Ott 1993).

In term of $N \times N$ matrix, the RPs can be defined as:

$$RPs_{i,j} = \Theta(\varepsilon_i - \|\vec{x_i} - \vec{x_j}\|). \tag{18}$$

The $\varepsilon_i$ is a predefined cut-off distance, $\|.\|$ is a norm (i.e Euclidean norm) and $\Theta(x)$ is the Heaviside function. It is worth noting that the matrix takes the form of zero (or black color) and one (or white color), respectively. The advantage of RPs suggested by Marwan over other recurrences in literature is the shorter and even non-stationary data which gives precise information on the dynamic state of the systems (Kac 1947; Balakrishnan *et al.* 2000). Based on the method suggested by Schinkel *et al.* (2008), the choice of $0.04d_A \le \varepsilon \le 0.7d_A$ was used in this work. That is, the recurrence threshold was chosen to range from 4% - 7% of the maximum attractor diameter (i.e., 8% - 14% of the corresponding attractor radius), where $\varepsilon$ is the recurrence threshold, and dA is the maximum attractor diameter. The method of finding the neighbors of the phase space trajectory is the euclidean norm between normalized vectors and was used throughout this study. The data used was also normalized to zero mean and standard deviation of one. However, the RQA is based largely on the distribution of the length of the diagonal structures of RP. In other to quantify RPs, Webber and Zbilut (1994) proposed an RQA extension which reveals several measures based on diagonally oriented lines such as recurrence point density and diagonal structures such as determinism, divergence (i.e the inverse of the maximal length of diagonal structures), entropy, trend (or drift) and the recurrence rate. The RQP based on determinism (DET), Linemax (Lmax) and entropy (ENT) measures were used for this research.

(i)Determinism (DET is the ratio of recurrence points forming diagonal structures to all recurrence points.

$$DET = \frac{\sum_{l=l_{min}}^{N} lP^{\varepsilon}(l)}{\sum_{ij}^{N} R_{ij}} \tag{19}$$

where $\varepsilon$ is the threshold, $P^{\varepsilon}(l)$ is the histogram of the length $l$ of the diagonal structures. For $l_{min} = 1$, then the determinism (DET) is equal to recurrence rate (RR). Periodic signals (e.g., sine waves) will give very long diagonal lines. Chaotic signals (e.g. Henon attractor) will give very short diagonal lines, and stochastic signals (e.g., random numbers) will give no diagonal lines (Shannon 1948). This has been used to quantify how deterministic a system is (Webber and Zbilut 1994).





(ii) Entropy (ENT), that is, the Shannon information entropy (Trulla *et al.* 1986), is a measure
of signal complexity. It shows the richness of deterministic structuring. However, entropy depends
sensitively on the bin number and therefore different for realization of the same process and data
preparation (Marwan *et al.* 2007)

$$ENT = -\sum_{l=l_{min}}^{N} P(l)lnP(l), \ \ P(l) = \frac{P^{\varepsilon}(l)}{\sum_{l=l_{min}}^{N} P^{\varepsilon}(l)} \tag{20}$$

(iii)Linemax (LMAX) is defined as the length of the longest diagonal line segment in the plot,
excluding the main diagonal line of identity. This particular variable is important as it is related to
the Largest possible Lyapunov Exponent (Thiel *et al.* 2002). The shorter the longest line is, the more
divergent the trajectories will be. A periodic signal will give long line segments, while short lines
indicate chaos.

$$L_{max} = max(\{l_i; i = 1, ...N_l\}), \ \ DIV = \frac{1}{L_{max}} \tag{21}$$

## 4   Results and Discussion

The global solar radiation has been estimated using the least square regression. The combined me-
teorological parameters used in this work may add to the accuracy of the global solar irradiance
prediction model (1). The properties of estimates a, b, c and n shown in table 2 confirm the signif-
icance of the estimates at 0.05 i.e. $\alpha = 0.05$ which is equivalent to a 95 percent significant level. It
can be observed from the that the values of estimates a, b, c, and n were uniquely different from one
station to another showing the relationship between the solar ratio ($R_s$) with relative humidity ($RH$)
and exponent of air temperature difference. The values of an estimate, $a$, which is the intercept of the
nonlinear combination of solar ratio, relative humidity and temperature difference ($TD$) were all pos-
itive with the highest value at Akure (Coastal zone) and Yola (Guinea savannah zone), respectively.
The least values were observed in Ilorin (Midland zone) and Kano (Sahel savannah zone). Meanwhile,
the values of estimates $b$ and $c$ were all negative and positive respectively in all the study locations.
These indicate that relative humidity which has negative estimate b shows decreasing trends with
increasing intensity of global solar radiation i.e. they have an inverse relationship with each other. On
the other hand, the exponent temperature difference which has positive estimate c shows increasing
trends with increasing intensity of global solar radiation i.e. they have a direct relationship with each
other.

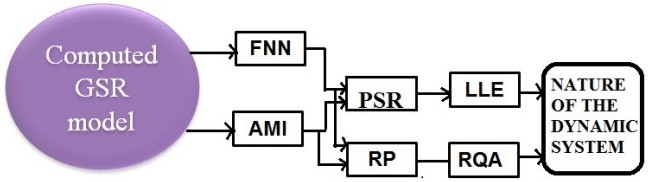

Figure 1: Typical block diagram for Chaotic signature Analyses


Table 2: Parameter estimates and their properties

| Station | Model Parameter Estimates | | | | Properties of Estimates | | |
|---|---|---|---|---|---|---|---|
| | a | b | c | n | SSE | $R^2$ | RMSE |
| Kano | 1.422 | -0.008282 | 2.815 | 0.9285 | 3.500 | 0.9184 | 0.01787 |
| Yola | 9.155 | -0.006934 | 0.08652 | 0.3400 | 14.22 | 0.6683 | 0.03603 |
| Ilorin | 1.37 | -0.007249 | 0.9677 | 0.5100 | 14.57 | 0.6602 | 0.03647 |
| Akure | 74.26 | -0.006797 | 0.009842 | 0.3270 | 14.22 | 0.6684 | 0.03603 |

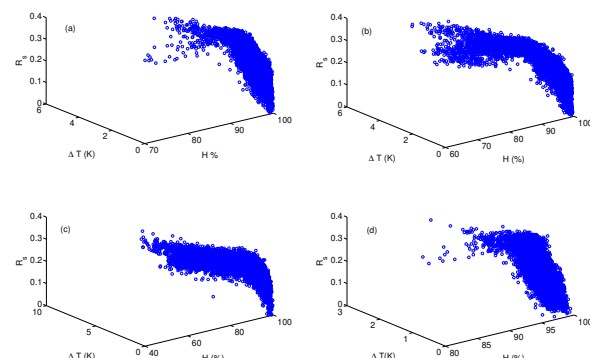

Figure 2: The 3D-Phase portrait Global Solar radiation model relating solar radiation, relative humidity and temperature difference for (a) Akure, (b) Kano, (c) Ilorin and (d) Yola

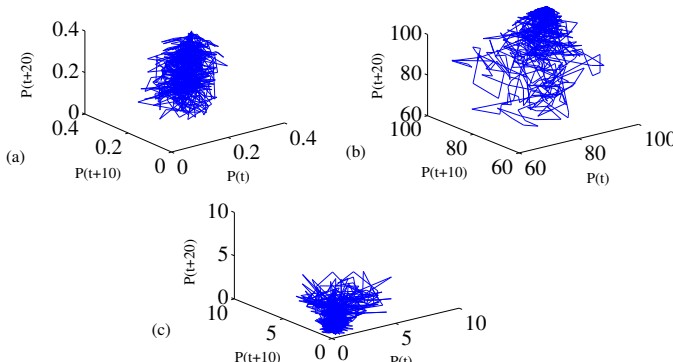

Figure 3: Typical 3D-Phase space reconstruction for the (a) Solar radiation, (b) Relative humidity and (c) Temperature difference



$$RS \Longleftrightarrow RH$$

$$RS \Longleftrightarrow TD$$

Figure 4: Daily Variation of Solar Radiation with Relative Humidity and Temperature Difference in (a) Akure, (b) Kano, (c) Ilorin and (d) Yola



The block diagram revealing the breakdown analyses of the chaotic measures is presented in Fig.(1). We also consider the 3D-Phase portrait, which reveals the relationship between the three meteorological parameters as plotted in Fig. (2). The phase space construction which represents the state of any real-world systems taking into consideration the dynamics emanating from the set of its state variable is plotted in Fig.(3). The PSR for the four regions between 2005-2016 were plotted using equation (14). The phase plots motions exhibit random-like and concentrated points at the center which indicates evidence of chaos. The average daily variations of solar ratio with the relative humidity in Akure and Yola has similar patterns, and also that of Kano and Ilorin (see Figs.4(a) and d). It can be observed that solar radiation has higher values in the first 50 days of the year and in the last 60 days of the year corresponding to the month of January, February, November, and December. Conversely, the values of the relative humidity were very low during these periods. Meanwhile, the average daily variations solar ratio with the differences between minimum and maximum temperatures in Akure, Kano, Ilorin, and Yola were also captured in Figs. 4(a-d). The patterns of the variability were similar in all the locations. These two climate variables vary directly with each other indicating that the degree of temperature in a particular area is a function of the intensity of solar irradiance of such area. They have peak values during the dry season days and minimum values during the core raining days as it had been established in the literature. It can also be observed from the figures that the temperature difference was all positive across the four locations. This indicates that the surface temperature of these locations is increasing and it is a signal to the possibility of global warming due to climate change. The positive differences may be attributed to the high population density and other anthropogenic activities on the land use cover associated with these locations. The RMSE values were very low and the value of $R^2$ is very closed to 1, which depicts a good prediction for all locations. However, highest values of both SSE and RMSE, as well as the lowest value of $R^2$ were observed in kano station, which indicates high solar irradiance location. Meanwhile, close values of SSE and RMSE, but a high percentage of $R^2$ were observed in Yola, Ilorin and Akure stations.

Also, the typical plots for the false nearest neighbors against embedded dimension(m) and the AMI against delay, $\tau$ were then plotted as shown in Fig.5(a). The choice of embedded dimension, m=7, and delay time, $\tau = 10$, is essential for phase PSR in this work to avoid over embedded (Fraser and Swinney 1986; Kennel *et al.* 1992). However, the choice of $\tau \geq 7$ and $m \geq 5$ values of delay and embedding dimension, respectively, are suitable for the analysis of the given data for all stations. The number of FNN plotted against the embedded dimension depicts the variation of solar ratio and the temperature difference in each station which is very high and similar compared to the low values of the relative humidity in each station (see Fig.5(b)). The PSR preserves relevant geometrical and dynamical invariants, such as the fractal dimensions of the attractor, the entropies, or the Lyapunov exponents (Fraser and Swinney 1986; Kennel *et al.* 1992).

The positive maximum Lyapunov exponent can be observed for solar radiation in Midland, Coastal and Guinnea Savannah regions (see Fig.6(a)), while Fig.6(b) shows higher positive values of Lyapunov exponent through-out the Sahel Savannah region. The direct relationship between Lyapunov exponent and Tsallis entropy has been perfectly displayed in Fig. 7(a) for Coastal region and Fig.7(b) for Sahel savannah zone, respectively. Also, the direct relationship and indirect relationship among the complex meteorological parameters have been revealed, which indicates their availability through-out the whole year. The monthly and seasonal underlying time series dynamics can also be investigated further using RQs. To avoid over embedding, the choice of m=7, $\tau = 10$ were used to characterize the abrupt behavior for the dry months of the year when solar irradiation has been confirmed to be higher at the Coastal, Midland and Guinea Savanna zones. However, behaviors but more intensive solar irradiation for a long period of months can also be noticed for the Sahel Savannah zone. The prediction at low threshold frequency ($\varepsilon$) of $0.7\sigma$ ($\sigma$ is the normalized distance euclidean norm) leads to more clusters of recurrence points around the diagonal lines when observed during the dry and wet seasons of the year. The transition from black to white bands marks the transition in the process i.e more chaotic or more complex in either solar radiation and relative humidity during those periods





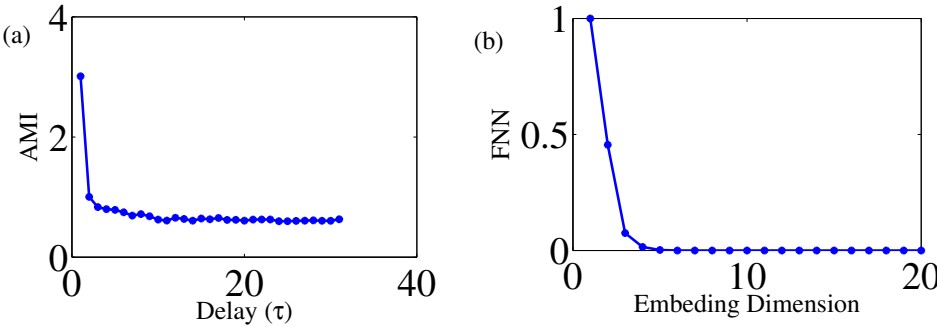

Figure 5: Typical (a) AMI plotted against time delay, and (b) False nearest neighbor for the selected location

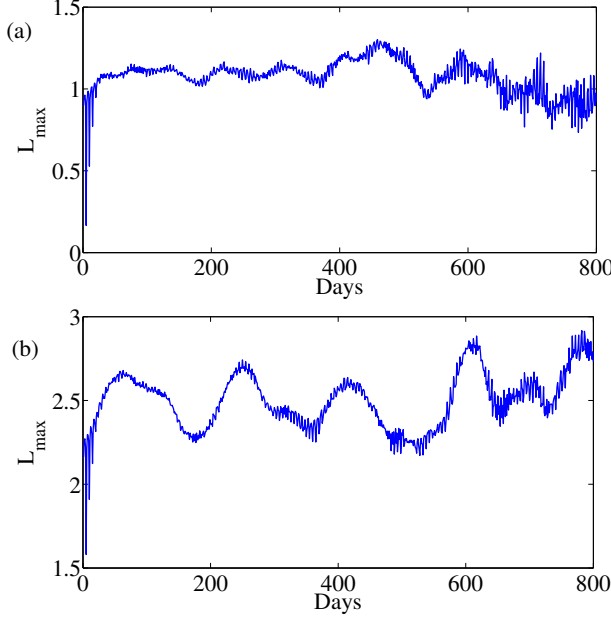

Figure 6: Typical (a) $L_{max}$ for Coastal region, and (b) $L_{max}$ for Sahel Savannah region





(see Fig. 8). Coastal regions reveal the presence of checkerboard structures around the diagonal lines. For instance, high insolation has been noticed at days < 100, 300 ≤days≤ 500 and 700 ≤days≤ 900 for the dry seasons (see Fig.8(a)). Conversely, in Fig. 8(b), relative humidity shows inverse trend during these insolation periods of the year, but high during the wet season of the year. Similar events can also be observed in Kano station (see Fig.9(a-b)), however, more solar radiation is observed during this period of the year which can be a result of harmattan and the evidence of chaos from the oscillatory nature of the system. Fig.9(b)shows little rain throughout the year.

Besides, RQA for seasonal influence on the dry season, the wet season and transition periods were also considered in this work. To commensurate the aforementioned behaviors, the occurrences such as August break i.e. temporary cessation of rain within a few days, on-set and off-set of both the dry season and rainy season months normally referred to as transition periods (i.e. March, April, October, and November), rainy season months (i.e. May and September) and dry season (i.e. November, December, January, and February) open ground for more information on the underlying dynamics of the climatological parameters. It is obvious in our result that the complex signature can be observed throughout the year for all the available parameters, especially during the peak periods of the year. For coastal region (see Figs.10(a)), only one long dry season, i.e. November to June, and short dry periods of the month (i.e. August break) can be observed in one year. However, the value of linemax (Lmax), determinism (DET), and entropy (ENT) for solar radiation can be seen to be very high at the beginning of each year (i.e January and February) with a gradual decrease towards the middle of the year before rising again at the return of the dry season (i.e. November and December). In contrast, the Sahel Savannah region experience very long but one dry season months for the year, usually from November of one year to May of the following year (see Fig. 10(b)). The chaotic signatures that were discovered through the aforementioned oscillatory nature of both solar radiation and relative humidity clearly show evidence of chaos, that is, higher (dry/wet) and lower (wet/dry) chaoticity, respectively during the season months of the year. The complex signatures across the regions also confirm the direct relationship of a temperature difference to solar radiation for the period under study. However, daily or monthly variation in solar irradiance which is a function of the aforementioned meteorological parameters may be attributed to the effect of the intertropical discontinuity, which affects the atmospheric stability during these periods. The results also indicate the availability of solar radiation throughout the whole year, which is an associated factor for the tropical regions.

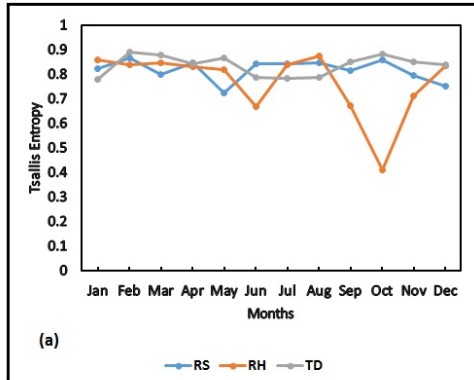
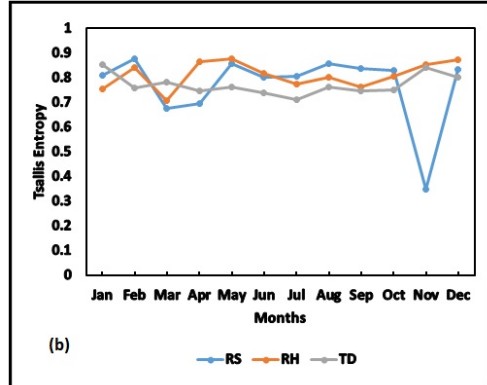

Figure 7: Typical (a) Tsallis Entropy for Coastal region, and (b) Tsallis Entropy for Sahel Savannah region

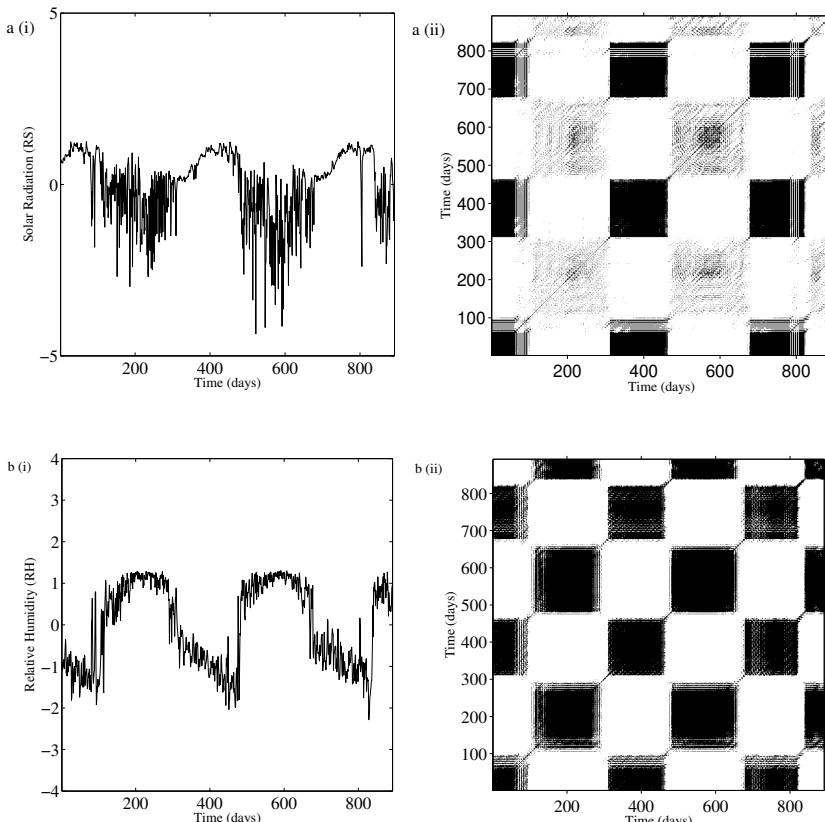

Figure 8: Underlying dynamics for Akure station showing a(i)time series for RS, b(i) time series for RH, a(ii) recurrrence plot for RS and b(ii) recurrence plot for RH; dimension: 7, delay: 10, threshold: 0.7 $\sigma$ (normalized distance euclidean norm)

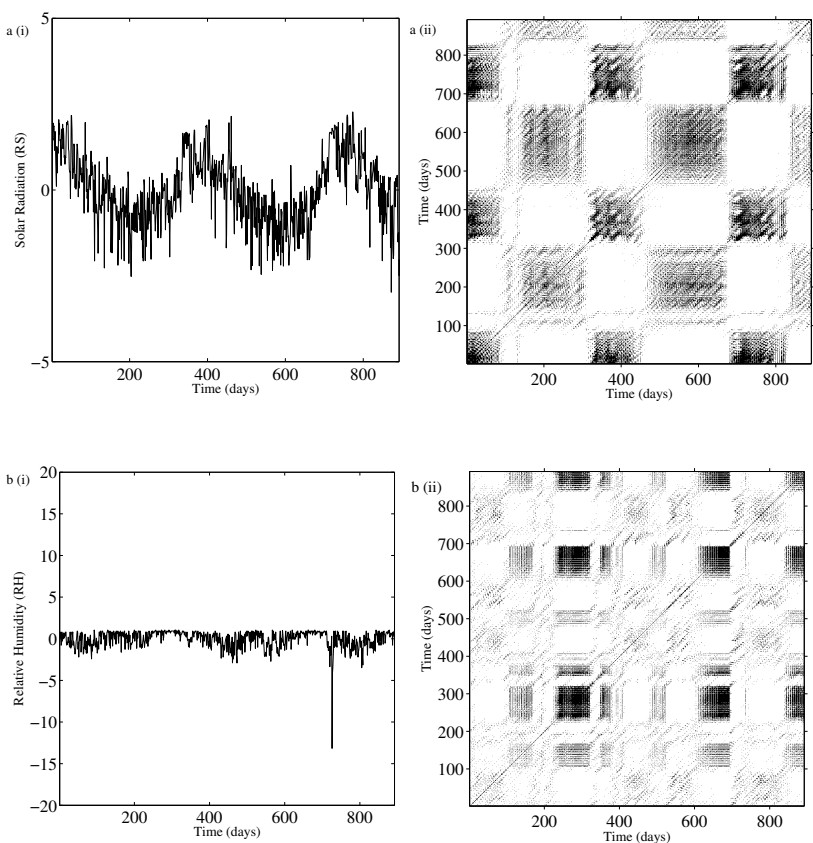

Figure 9: Underlying dynamics for Kano station showing a(i)time series for RS, b(i) time series for RH, a(ii) recurrrence plot for RS and b(ii) recurrence plot for RH; dimension: 7, delay: 10, threshold: 0.7 $\sigma$ (normalized distance euclidean norm)


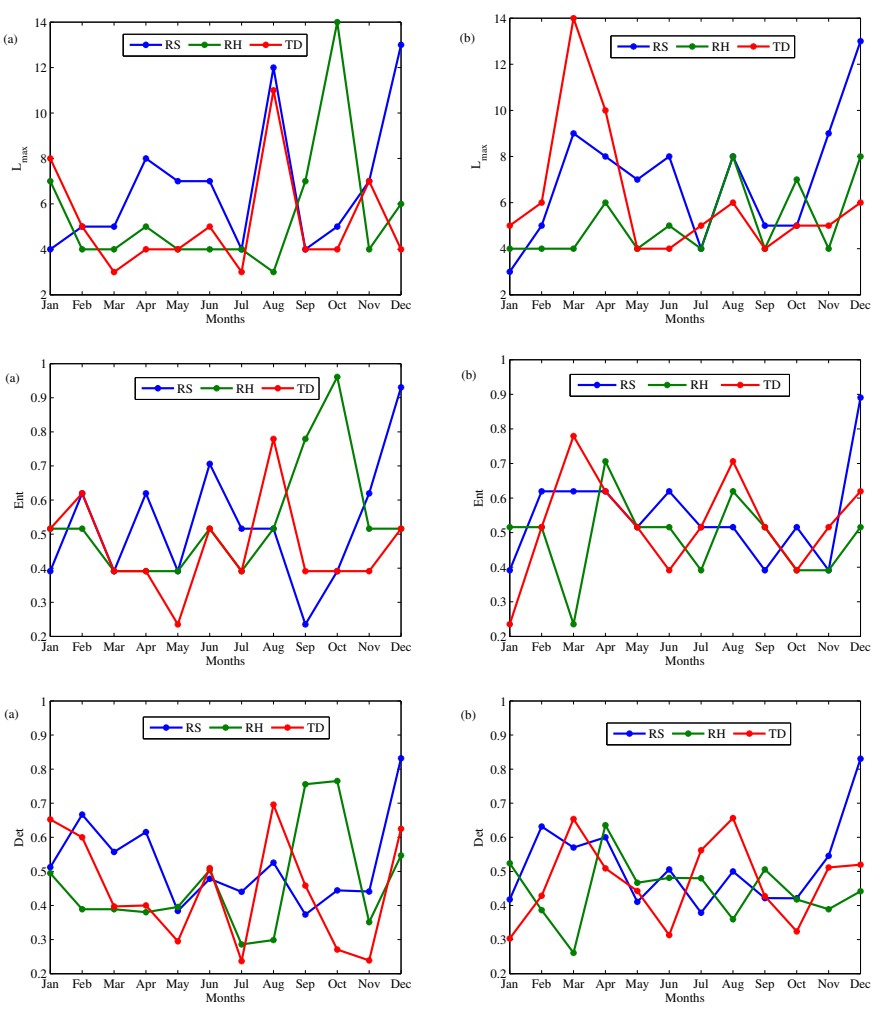

Figure 10: Lmax, Entropy and Determinism for (a)Akure and (b)Kano stations




The results herein, confirmed that the values of solar radiation were very low throughout June, July and August. This may be attributed to the presence of cloud cover as a result of rainfall which is normally prevalence during those periods. Although the patterns of variations between the solar radiation and relative humidity in Kano and Ilorin were oscillatory, yet the inverse relation and behavior between them were vividly maintained. The inverse and complex signature attributes can also be noticed in Akure and Yola stations. Therefore, it can asserted that the intensity of solar radiation can be greatly attenuated by the increase and decrease of the relative humidity. Meanwhile, the temperature difference across the four regions also ascertained the complexity which is direct to solar radiation strength. In summary, the global solar insolation measures have been based on the diagonal lines of RP. The choice of embedding is paramount in estimating the aforementioned measures. In general, when the meteorological parameters of a natural system are chaotic, it implies that the Lmax, determinism and entropy are all low which depicts that the system is far from equilibrium. Therefore, the similarities are based on the fact that their measurements are based on the diagonal lines of RP which are clearly stated in equations (22), (23) and (24).

# 5 Conclusion

Predicting and regular studying of underlying dynamics of global solar irradiance data is very important for solar inverter designers in Nigeria, a tropic region. Thus, in this work, we propose a new nonlinear Solar radiation model and have been evaluated using statistical tools $R^2$, RMSE, and SSE. Relative humidity and temperature difference have been identified as a perfect climatological parameters to predict global solar irradiance. The observed estimates vary across each region which indicates the appropriate relationship between the meteorological parameters. The complexity or degree of solar irradiance in relations to relative humidity and temperature difference across the selected regions were reported. The choice of m=10 and $\tau = 7$ to avoid over-embedded has been chosen, which form the bases for other studies. The observed clusters points around the parallel diagonal lines with short segments, which implies the presence of chaos. Additional complex measure, the RQA also shows that the solar radiation during the dry season of the months has lower values of Lmax, determinism and entropy and higher values during the wet season of the months. The negative estimate of relative humidity commensurates the decreasing trends with increasing global solar radiation, unlike the direct relationship that was observed between global solar radiation and the temperature difference for each region. However, the availability of solar irradiance in all selected zones has been discovered with the highest irradiance observed in Sahel savanna zone, which shows that the zone exhibit more complex solar irradiance than the other selected regions. Other regions such as Coastal, Midland and Guinea savannah regions were suitable for Agricultural purposes during the wet season as well as solar energy capture through-out the year. Therefore, with the implementation of accurate and efficient prediction, we will be able to identify which regions is/are suitable for optimal capture of solar radiation for human use and energy source designing.

# 6 Availability of data

The data that support the findings of this study are available from the corresponding author upon reasonable request and can also be obtained from the archive of ERA-Interim.

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
