# Peer review of "Chaotic Signatures and Global Solar Radiation model estimate over Nigeria, a Tropical region"

_Nonlinear Processes in Geophysics, 2020_

## Referee Comment (RC1) · Anonymous Referee #1 · 24 Oct 2020

The manuscript investigates the use of a reconstruction model for daily solar global radiation based on temperature and relative humidity over 4 locations in Nigeria.

The topic of the manuscript is of great interests given the possible applications of such a model in the region. The merit of the paper is to study the application of the model through advanced mathematical techniques. The methodology applied is not original and poorly introduced by the authors, nonetheless the information derived can potentially be useful for other researchers working on the same topic over the same region or in other parts of the world. On the other hand, the manuscript in its present form has too many weaknesses for being considered for publication. It is my opinion that

the author should completely revise the manuscript and re-submit it for another review round. In its present form, the manuscript is not suitable for publication.

The most significant weakness is the organization of the manuscript. The Introduction is a mere review of the scientific literature, without mentioning authors motivations and highlighting the original part of this work. The method is very poorly presented. The scores used in the evaluation are not properly introduced and I suspect the equations defining them are wrong. The section on chaotic quantifiers comes as a surprise, since it is quite demanding from a mathematical point of view and poorly introduced, it is so disconnected from the rest of the paper that I was left wondering "Why do they do that?" for all the time I was reading it. The discussion of results is not easy to follow, given all the weaknesses reported above. The conclusions are very hard to follow, since statements like "the availability of solar irradiance in all selected zones has been discovered with the highest irradiance observed in Sahel savanna zone, which shows that the zone exhibit more complex solar irradiance than the other selected regions." I can not really understand them. The authors should make an effort to better communicate their findings. The way the study is communicated leaves much to be desired. Specific comments follow.

I hope the author would be willing to spend some more time rewriting the manuscript, such that this interesting work will get the attention it deserves.

Comments:

Title: remove "a Tropical region". "Over Nigeria" is fine.

Abstract: revise the text. Abbreviations are used without explaining their meaning (e.g. PV, RPs, RQA), you should explain them in the text and not only as Keywords. You should not use phrases like "the well-known statistical tools" without introducing them. The reader should not be left to guess the meaning of your statements. Try to be more specific.

Introduction: good review of scientific literature. The concise introduction on your own research must be rewritten. For example, by reading your introduction I'm not sure whether you have developed a statistical model for spatial analysis of solar global radiation or a model to forecast it. You should be more specific about your motivations, implementation choices (e.g. why not use solar radiation from reanalysis directly?) and methods.

Page 2. Lines 12-14. "The source [solar energy] has been capable of reducing climate and weather events". Not sure what this means. Please rephrase it or delete.

Page 2. Lines 21-23. I think here you are mixing processes that are very different from each other, giving the impression that they are all the same process. Statistical techniques are not methods to collect data. Statistical techniques can be used to estimate values of an observable quantity between locations where these values are known from other processes (e.g. observations, models). You should make it clear that there are measurements and models. They both return estimates of a quantity, though with very different properties.

Page 2. Lines 34-35. "various atmospheric situations have been applied to the field of science and engineering." the meaning is not clear.

Page 2. Lines 37-48 I do not get your point here "the complexity of the global solar radiation and the inherent irregularities occurrence in space can be identified to be chaotic or hyperchaotic based on the availability of climatological data which has been enriched with the theory of nonlinear dynamics." What do you mean when you state that global solar radiation is chaotic? Are you thinking at the forecasts and the fact that is dependent on initial conditions? The text jumps from saying that solar global radiation is highly variable in space to defining it as "chaotic", a definition of the term is needed here.

Page 2. Line 44. "And revealed." Please continue the statement

Page 3. Lines 28-31. Please check the units (W/m^2 instead of W-Delta-m^2). Add the units to E0.

Sec. 2.1. This is the core of your paper. Here you present your model for the reconstruction of solar global radiation, given daily temperature and humidity. However, the title of this section is "study area and data analysis", which do not even mention the fact that here you are presenting your model. In addition, the reader gets the impression that Eqs.(1)-(2) have not been used before, which is not true. Please, specify your references for these equations. How did you get the idea to use them? Why not use directly the solar radiation one can derive from reanalysis? I can not answer this question, because you should have better specified in the Introduction your motivations. I think your objective is to estimate solar global radiation from a very reduced set of meteorological parameters (temperature and relative humidity), which are commonly measured by weather stations. In this way, it would not be necessary to install several expensive instruments in the regions to measure solar radiation, since it could be extracted from less expensive measurements. If this is your objective, then state it clearly in the Introduction.

Page 4. Section 2.2. Define the meanings of the abbreviations RMSE, SSE and R^2. I guess that they are:

- RMSE: root mean squared error, even though your definition in Eq.(9) multiplies the conventional RMSE by 100. Why so? In addition, the summation under the square root should have an index.

- SSE: it seems that Eq.(10) is a weird definition of a cumulation of squared deviations. Is this really needed, since you have defined RMSE. Why the two summations?

- R^2: I think this is the mean‐squared error skill score (MSESS) and not a correlation. See for example: Isotta, F. A., Begert, M., & Frei, C. (2019). Long‐term consistent monthly temperature and precipitation grid data sets for Switzerland over the past 150 years. Journal of Geophysical Research: Atmospheres, 124, 3783–3799.

https://doi.org/10.1029/2018JD029910

As all skill scores, it is positively oriented with the perfect score being 1. Please correct the text.

Section 3. The whole purpose of this section is obscure to me. I understand that the authors introduce advanced mathematical tools as diagnostics of the model they have presented in sec. 2.1. Your model for solar radiation is not too complex. Please explain in plain word what are the benefits of conducting an analysis on the chaotic properties of the parameter space. How this analysis is used to improve your model presented in Sec. 2.1? Eq.(12). A new operator is introduced <...> without explanation.

Page 8. Line 13. "The global solar radiation has been estimated using the least square regression". This is not true and it creates confusion about your method. You are applying the method presented in Sec. 2.1. Why are you mentioning least square regression?

---

## Referee Comment (RC2) · Anonymous Referee #2 · 23 Nov 2020

Report on the paper "Chaotic Signatures and Global Solar Radiation model estimate over Nigeria, a Tropical region".

Trying to write a report on this manuscript is a tough task. The article is unclear, the style sloppy, and the authors do not bother to present it in a minimally attractive way. I do not dare say that it does not contain any relevant information, but if there is something which is worth being published it is well hidden.

- General considerations

The manuscript is much too long (21 pages, including the plots) for its scientific contents. There is a short abstract followed by a longer abstract (in boldface). Is this

second abstract included at the request of the editors?

The abstracts do not clarify anything and are full of acronyms, which renders the text difficult to read. Instead of prompting the reader to follow with interest what the authors want to convey, such overuse of acronyms makes the reader to look forward to reaching the end of the paragraph soon.

- Comments on the science

Take the title to start with: "Chaotic Signatures and Global Solar Radiation model estimate over Nigeria, a Tropical region".

"Chaotic Signatures" exhibited by what? What is the meaning of "Global Solar Radiation model estimate"? Maybe "a model for estimating global solar radiation" is meant?

It is not clear to me what is meant by "Global Solar Radiation". Why don't the authors define clearly the concept at the beginning of the article? Is here "global" the opposite of "local" or do they mean "global" in the sense that it is the sum of the diffuse and direct solar radiation reaching one horizontal square meter on the ground? They should make clear whether it is an incident energy flux density (W/mˆ2) or whether it is integrated over a time interval (J/mˆ2), for instance the daily insolation on a horizontal plane. I suggest that the concept of "global solar radiation" is succinctly and clearly presented at the beginning and the difference with "solar irradiance" is explained.

Regarding the scientific contents: If I have understood it correctly, the authors parametrize the "global solar radiation" and say that it depends on the extraterrestrial radiation corrected by a linear term (that takes account of the relative humidity) and an exponential term (that takes account of the difference between the maximum and the minimum temperatures). I think this is a sort of clearness index introduced by Liu and Jordan, which is used when the amount of cloud cover is not known, and it is not that clear that it works well (see. e.g., "Strengths and limitations of the Liu and Jordan model to determine diffuse from global irradiance" by LeBaron & Dirmhirn 1983

in "Solar Energy". vol. 31). I do not understand why the "global solar radiation" should depend on the difference between the maximum and the minimum daily temperatures (I do not say it is wrong, but some explanation should have been provided).

The authors leave three constants {a, b, c} that are determined through a least-squares fitting and then they say that the correlation is good. It is not clear to me whether the correlation has been obtained using a time series different from the one they used to determine the constants of the model. Furthermore, it is not even clear to me that they have performed these correlations based on observations. Maybe everything is OK and I am confused here, but the text is quite unclear and it takes a huge effort to understand it.

The general impression I get from the manuscript is that it is one of those papers on data analysis where the authors have a data processing program and a data set at their disposal. The program is fed with data from one end and something comes out at the other end.

Maybe there is something interesting hidden in the long, sloppy manuscript, but it is the author's task to present it in an attractive way by means of a fluent, concise and logically structured text; it is not the referee's task to decipher it.

- Style and writing

Is it necessary to emphasize in the title that Nigeria is a tropical region? This should be obvious to any reader of this journal, but in case the authors feel that this information is necessary, it could be said later in the manuscript.

Many sentences are trivialities or fillers. For example, in the Introduction (page 2, line 34), the Lorenz model of atmospheric convection is mentioned, but it has nothing to do with the contents of the paper and it is not used later. It makes the impression that, in order to justify that they intend to publish their manuscript in a journal on "non-linear processes", the authors had to mention one of the seminal papers in non-linear

dynamics, be it used later or not.

An instance of useless text [page 1, lines 19, 20, 21]: "The information, therefore, suggests how vital the solar irradiance can be useful in Agriculture and Photovoltaic technology companies".

Another instance of poor writing [page 1, lines 8, 9, 10]: "The well-known statistical tools were used to analyze the chosen meteorological parameters and the correlation was found to be perfect, (...)". Here the article "the" is wrong (the meaning of the sentence is changed). It turns out that the correlation is not only good, but perfect... the correlation of what with what?

Why do some substantives and adjectives start with a capital letter ("Signatures", "Tropical") but others don't? Why "Photovoltaic technology companies". Why "Relative Humidity" (uppercase) but "temperature" (lowercase) in page 3 lines 23 and 24? In page 3 line 28 the "solar constant" (lowercase, while they write "Global Solar Radiation") is written as 1353 W Delta m$^2$ (I assume the capital Greek letter delta resulted by mistake from a slash /). Somebody should have taken care of such details before submitting the manuscript.

Apart from the poor writing there are grammatical and orthographic errors. The text should have been revised by someone who is fluent in English before sending it to the referees.

Recommendation to the editor: Reject the manuscript.

---

## Author Comment (AC1) · 28 Dec 2020

Short Response on RC1: The manuscript investigates the use of a reconstruction model for daily solar global radiation based on temperature and relative humidity over 4 locations in Nigeria. The topic of the manuscript is of great interests given the possible applications of such a model in the region. The merit of the paper is to study the application of the model through advanced mathematical techniques. The authors appreciate the positive assessment of the paper. Necessary corrections and adjustments to the present form have been done. The main aim of the paper is to determine the variation of chaos in solar radiation data in different climatic conditions of Nigeria
which has been included. Chaos theory offers an insightful approach to investigating the global solar radiation behavior of time series data without assumptions about the underlying distribution. The form of the paper is now clear with precise information and new findings are clearly stated. Suggested additional information has been taking care of in the paper, while the full explanation of the abbreviations has been included in the manuscript.

Short Response on RC2: The authors also appreciate the positive assessment of the paper. As discussed above, necessary corrections and adjustments to the present form have been effected. The main aim of the paper is to determine the variation of chaos in solar radiation data in different climatic conditions of Nigeria which has been included. Chaos theory offers an insightful approach to investigating the global solar radiation behavior of time series data without assumptions about the underlying distribution. Authors motivations and highlighting the original part of the work have been included in the reviewed version of the paper. Complete revision on the introduction and the body content of the paper has been done.

---

## Editor Comment (EC1) · Vicente Perez-Munuzuri (Editor) · 6 Jan 2021

Dear authors. Thanks for your comments to the referee reports. However, the journal establishes the need to give a detailed answer to the referees, otherwise we could not continue with the editorial process of your paper. Thanks and Happy New Year!